# Occurrence and characteristics of extended-spectrum-β-lactamase- and pAmpC-producing *Klebsiella pneumoniae* isolated from companion animals with urinary tract infections

Megan Min Yi Lee[1], Nan-Ling Kuan[1,2], Zhi-Yi Li[1], Kuang-Sheng Yeh[1]*

1 Department of Veterinary Medicine, School of Veterinary Medicine, National Taiwan University, Taipei, Taiwan, 2 Biology Division, Veterinary Research Institute, Ministry of Agriculture, Tansui, New Taipei City, Taiwan

* ksyeh@ntu.edu.tw

**Data Availability Statement:** All relevant data are within the paper and its supporting information files.

## Abstract

This study examined 70 *Klebsiella pneumoniae* isolates derived from companion animals with urinary tract infections in Taiwan. Overall, 81% (57/70) of the isolates carried extended-spectrum β-lactamase (ESBL) and/or plasmid-encoded AmpC (pAmpC) genes. ESBL genes were detected in 19 samples, with $bla_{CTX-M-1}$, $bla_{CTX-M-9}$, and $bla_{SHV}$ being the predominant groups. pAmpC genes were detected in 56 isolates, with $bla_{CIT}$ and $bla_{DHA}$ being the predominant groups. Multilocus sequence typing revealed that sequence types (ST)11, ST15, and ST655 were prevalent. *wabG*, *uge*, *entB*, *mrkD*, and *fimH* were identified as primary virulence genes. Two isolates demonstrated a hypermucoviscosity phenotype in the string test. Antimicrobial susceptibility testing exhibited high resistance to β-lactams and fluoroquinolones in ESBL-positive isolates but low resistance to aminoglycosides, sulfonamides, and carbapenems. Isolates carrying pAmpC genes exhibited resistance to penicillin-class β-lactams. These findings provide valuable insights into the role of *K. pneumoniae* in the context of the concept of One Health.

## Introduction

*Klebsiella pneumoniae* is a Gram-negative, nonmotile, encapsulated bacterium belonging to the *Enterobacteriaceae* family. This bacterium is a crucial opportunistic pathogen that causes infections in humans, particularly in those with a compromised immune system and urinary tract infections (UTIs) [1]. The emergence and spread of multidrug-resistant *K. pneumoniae* strains pose a challenge for clinical treatment [2]. The World Health Organization has recognized *K. pneumoniae* and other *Enterobacteriaceae* resistant to third-generation cephalosporins (3GCs) as a Priority 1 group [3]. 3GCs are commonly used for the treatment of UTIs [4, 5]. Moreover, *K. pneumoniae* is a common cause of UTIs in pets, with 14% of dogs and 3%–

**Funding:** This work was supported by a grant (NSTC-112-2313-B-002-031) from the National Science and Technology Council, Taiwan to K.-S. Yeh..The funders had no role in study design, data collection and analysis, decision to publish, or preparation of the manuscript.

**Competing interests:** The authors have declared that no competing interests exist.

19% of cats estimated to have a UTI during their lifetime [6–9]. *K. pneumoniae* also causes infections at other sites and can become resistant to 3GCs by producing extended-spectrum beta-lactamases (ESBLs) and AmpC β-lactamases. ESBL genes are mainly located on plasmids of various molecular sizes, which enhance their horizontal transfer, in addition, those plasmids can confer resistance to other antibiotics such as fluoroquinolones and aminoglycosides [10]. AmpC β-lactamases can also be located on plasmids or chromosomally encoded [11, 12].

In clinical settings, *K. pneumoniae* strains carrying pAmpC are commonly observed to harbor ESBL. These two types of β-lactamase genes can be located on the same plasmid or on different plasmids [13]. In addition to antibiotic resistance, *K. pneumoniae* has four major virulence factors: fimbriae, capsules, lipopolysaccharides, and iron-acquiring proteins [14]. Several other factors such as OMPs, porins, efflux pumps, iron transport systems, and genes involved in allantoin metabolism have also been described, but some of their function remained to be elucidated [15]. Briefly, *fimH* and *mrkD* encode the adhesion molecules of type 1 and type 3 fimbriae, respectively. Fimbrial structures mediate *K. pneumoniae* binding to host epithelial cells. RmpA is a plasmid-located virulence factor of *K. pneumoniae* that regulates the synthesis of capsular polysaccharides. LPS production is regulated by the uridine diphosphate galacturonate 4-epimerase (*uge*) gene. Without this gene, *K. pneumoniae* is less able to cause urinary tract infections, pneumonia and sepsis. The *Klebsiella* ferric iron uptake (*kfu*) is a regulator gene of the iron transport system involved in iron absorption. Siderophores are small molecules that can compete iron from the host's iron-chelating proteins. Siderophores expressed in *K. pneumoniae* include enterobactin, yersiniabactin and aerobactin. The gene associated with allantoin metabolism (*allS*) is used by bacteria to obtain carbon and nitrogen from the environment. Some of the genes encoding the aforementioned virulence factors are widely used to evaluate the pathogenicity of *K. pneumoniae* [16].

On the basis of differences in the accessory genome, *K. pneumoniae* can be divided into opportunistic, hypervirulent, and multidrug-resistant types [17]. Opportunistic *K. pneumoniae* still commonly causes hospital-acquired infections and is more likely to infect patients with compromised immune systems. Hypervirulent *K. pneumoniae* can infect individuals with healthy immune systems, causing various diseases, such as liver abscess, endophthalmitis, meningitis, and septic arthritis, primarily in community-acquired cases. The antibiotic resistance genes of multidrug-resistant *K. pneumoniae* are often located on transmissible plasmids. Some strains can accumulate more resistance genes through transmission and conjugation, forming extremely drug-resistant strains characterized by a super resistome [18, 19]. In conclusion, *K. pneumoniae* has evolved beyond merely being an opportunistic pathogen and now exhibits a diverse range of pathogenic behaviors

In contrast to wild animals and humans, pets and humans come in close contact over the long term, creating a relatively high chance for bacterial transmission between them [20]. Some ESBL-producing *K. pneumoniae* strains isolated from pets have been identified as high-risk bacterial populations that commonly infect humans [21–23]. This finding underscores the importance of *K. pneumoniae* in zoonotic transmission and public health [24]. According to the concept of One Health, this study focused on *K. pneumoniae* isolated from pets with UTIs. We determined the proportion of *K. pneumoniae* strains harboring ESBL, pAmpC, or both. The virulence factors of these resistant *K. pneumoniae* strains were examined through polymerase chain reaction (PCR). The relationships among these strains were analyzed using multilocus sequence typing (MLST). The findings of this study provide valuable information from a public health perspective.

## Materials and methods

### *K. pneumoniae* strain collection

Bacterial strains analyzed in this study were obtained from pet visits (only dogs and cats) at National Taiwan University Veterinary Hospital between January 1, 2014, and December 31, 2018. A total of 171 *K. pneumoniae* strains were isolated from dogs and cats, and 70 isolates were obtained from the urine samples of animals with UTIs. These collected bacteria were identified to the species level by using a Vitek-2 Compact microbial detection system (bioMérieux, Marcy I'Etoile, France) [25]. The bacterial strains were stored in Microbank System cryovials (Pro-Lab Diagnostics, Richmond Hill, ON, Canada) at −80˚C.

### Screening and confirmation of ESBL-producing *K. pneumoniae*

The *K. pneumoniae* strains were removed from the −80˚C freezer, inoculated on tryptone soy agar (Becton Dickison, Franklin Lakes, NJ, USA), and incubated at 37˚C for 16–18 hours. After confirming the growth of colonies, we inoculated them on ESBL selective culture medium, CHROMagar ESBL (CHROMagar, Paris, France), and incubated them at 37˚C for 16–18 hours for preliminary screening. On this culture medium, ESBL-producing *K. pneumoniae* yielded deep blue colonies, whereas non-ESBL-producing *K. pneumoniae* strains did not grow. The strains that screened positive in the initial tests were subjected to phenotypic confirmation. According to the Clinical and Laboratory Standards Institute guidelines, a phenotypic confirmatory test for ESBL production in *K. pneumoniae* can be performed using the disk diffusion method [26]. This method operates on the principle of inhibiting the activity of ESBL through β-lactamase clavulanic acid, enabling third-generation cephalosporins to kill the bacteria, thus broadening the inhibition zone. The difference in the size of the inhibition zone between the presence and absence of clavulanic acid was used to determine whether the strain produced ESBL. Briefly, the procedure involved aseptically collecting a bacterial colony by using a loop, mixing it with phosphate-buffered saline to adjust to a McFarland concentration of 0.5, and then uniformly applying it to Mueller–Hinton (MH) agar (Neogen, Lansing, MI, USA) using a sterile cotton swab. Four antibiotic discs, namely ceftazidime (CAZ, 30 μg), ceftazidime (30 μg)–clavulanic acid (10 μg, CAZ/CA), cefotaxime (CTX, 30 μg), and cefotaxime (30 μg)–clavulanic acid (10 μg, CTX/CA) were placed on the agar, and the plates were incubated at 37˚C for 16 to 18 hours. The discs were purchased from Becton Dickison. The difference in the size of inhibition zones served as the basis for interpretation. If the difference between CAZ/CA and CAZ was greater than or equal to 5 mm or the difference between CTX/CA and CTX was greater than or equal to 5 mm, the strain was considered to be an ESBL-positive strain. *K. pneumoniae* ATCC70063 and *E. coli* ATCC25922 served as positive and negative control groups for ESBL, respectively.

### DNA extraction

The DNA of the test strains was extracted through boiling [27]. Briefly, the tested *K. pneumoniae* strains were cultured for 16–18 h at 37˚C on tryptic soy agar plates (Becton Dickinson). A loopful of cells was collected and added to a microcentrifuge tube supplemented with 200 μL of double-distilled $H_2O$ (dd$H_2O$), and this mixture was boiled for 10 min. The supernatant was collected after centrifugation at 12,000 × *g* for 10 min and stored at −20˚C. This was used as the template in subsequent PCR experiments.

## β-lactamase genotype detection and sequencing analysis

PCR was performed to amplify target ESBL genes, including seven gene groups: $bla_{TEM}$, $bla_{SHV}$, and $bla_{CTX-M}$ (CTX-M-1, CTX-M-2, CTX-M-8, CTX-M-9, and CTX-M-25). The amplified genes were confirmed through DNA electrophoresis. The PCR mixture was prepared by combining 5 μL of DNA template, 1 μL each of 10 μM forward and reverse primers (Tri-I Biotech, New Taipei, Taiwan), 18 μL of sterile water, and 25 μL of 2× MasterMix (Ampliqon, Odense M, Denmark) in a 200-μL microcentrifuge tube, which was then placed in a PCR thermocycler (SensoQuest, Göttingen, Germany). The PCR conditions were as follows: an initial cycle for 5 min at 95˚C, followed by 35 cycles at 30 s each at 95˚C for denaturation, 40 s of annealing at a primer-determined temperature, and 60 s of extension at 72˚C, and finally a cycle at 72˚C for 10 min. The PCR products were stored at 4˚C. The product was examined through DNA electrophoresis to confirm ESBL gene amplification results. If the expected outcome was obtained, the product of the desired size was excised from the agarose gel. Then, the complete nucleic acid sequence was assembled using DNASTAR Lasergene-Seqman software (DANSTAR, Madison, WI, USA) based on forward and reverse primer sequences obtained from sequencing (Tri-I Biotech, Taipei, Taiwan). The results were uploaded to the β-lactamase database Beta-Lactamase DataBase (BLDB)-Structure and Function website (http://www.bldb.eu/) for β-lactamase genotype determination.

PCR was performed to detect commonly recognized pAmpC genes, namely $bla_{CIT}$, $bla_{DHA}$, $bla_{MOX}$, $bla_{EBC}$, and $bla_{FOX}$, following the same procedure as used for ESBL gene detection. The primers used to detect these β-lactamase genes are listed in Table 1. All 70 K. pneumoniae isolates were included in the PCR test.

## Antimicrobial susceptibility test

The disc agar diffusion method was used for antimicrobial susceptibility testing [26], and the selected drug discs were as follows: amoxicillin/clavulanate (20/10 μg), ampicillin (10 μg), augmentin (5 μg), cefixime (5 μg), cefotaxime (30 μg), cefovecin (30 μg), cefoxitin (30 μg), cefpodoxime (10 μg), cephalothin (30 μg), ciprofloxacin (30 μg), doxycycline (30 μg), enrofloxacin (5 μg), gentamicin (10 μg), imipenem (10 μg), tetracycline (30 μg), and trimethoprim/sulfamethoxazole (25 μg).

## Conjugation test

A conjugation test was conducted to determine whether the ESBL/pAmpC-producing K. pneumoniae strain isolated in the experiment could horizontally transfer its antibiotic resistance genes. Since 16 of the total 57 ESBL/pAmpC-producing K. pneumoniae strains were naturally resistant to sodium azide because they could grow on Mueller–Hinton agar supplemented with sodium azide for unknown reason; Therefore, these strains were excluded from the conjugation test. The number of strains used for the conjugation test was 41. The aforementioned K. pneumoniae strain was used as the donor, and E. coli J53 (ATCC BAA-2730TM), which is resistant to sodium azide, was used as the recipient. The two strains were co-cultivated and screened for successful transconjugants by using a double-selective MH agar medium containing cefotaxime (2 mg/L) and sodium azide (150 mg/L) (Sigma-Aldrid, Burlington, MA, USA) [34]. The presence of an antibiotic resistance gene in the transconjugant E. coli J53 was confirmed through PCR. The procedure involved the following steps: the donor K. pneumoniae strain and E. coli J53 strain were retrieved from frozen tubes and cultured in test tubes containing 5 mL of Luria–Bertani (LB) broth. The tubes were incubated at 37˚C for 16–18 hours, Subsequently, 0.5 mL of each bacterial suspension was mixed with 4.5 mL of fresh LB broth and incubated at 37˚C for 4 hours. Then, 0.5 mL of the donor solution and 0.5 mL of

**Table 1. Primers used to detect β-lactamase genes.**

| PCR target | Primer | Sequence (5'-3') | Annealing Tm (°C) | Predicted PCR size (bp) | Reference |
|---|---|---|---|---|---|
| $bla_{TEM}$ | TEM-F<br>TEM-R | TCGGGGAAATGTGCGCG<br>TGCTTAATCATGAGGCACC | 55 | 972 | [28] |
| $bla_{SHV}$ | SHV-F<br>SHV-R | GCCTTTATCGGCCCTCATCAA<br>TCCCGCAGATAAATCACCACAATG | 54 | 819 | [29] |
| $bla_{CTX-M-1}$ | CTX-M-1-F<br>CTX-M-1-R | CCCATGGTTAAAAAATCACTGC<br>CAGCGCTTTTGCCGTCTAAG | 54 | 942 | [30] |
| $bla_{CTX-M-2}$ | CTX-M-2-F<br>CTX-M-2-R | CGACGCTACCCCTGCTATT<br>CCAGCGTCAGATTTTTCAGG | 52 | 552 | [31] |
| $bla_{CTX-M-8}$ | CTX-M-8-F<br>CTX-M-8-R | TCGCGTTAAGCGGATGATGC<br>AACCCACGATGTGGGTAGC | 52 | 666 | [31] |
| $bla_{CTX-M-9}$ | CTX-M-9-F<br>CTX-M-9-R | ATGGTGACAAAGAGAGTGCAAC<br>TTACAGCCCTTCGGCGATGATT | 55 | 876 | [32] |
| $bla_{CTX-M-25}$ | CTX-M-25-F<br>CTX-M-25-R | GCACGATGACATTCGGG<br>AACCCACGATGTGGGTAGC | 52 | 327 | [31] |
| $bla_{pAmpC}$ | CIT-M-F<br>CIT-M-R | TGGCCAGAACTGACAGGCAAA<br>TTTCTCCTGAACGTCGCTGGC | 64 | 462 | [33] |
| $bla_{pAmpC}$ | MOX-M-F<br>MOX-M-R | GCTGCTCAAGGAGCACAGGAT<br>CACATTGACATAGGTGTGGTGC | 64 | 520 | [33] |
| $bla_{pAmpC}$ | DHA-M-F<br>DHA-M-R | AACTTTCACAGCTGTGCTGGGT<br>CCGTACGCATACTGGCTTTGC | 64 | 405 | [33] |
| $bla_{pAmpC}$ | EBC-M-F<br>EBC-M-R | TCGGTAAAGCCGATGTTGCGG<br>CTTCCACTGCGGCTGCCAGTT | 64 | 302 | [33] |
| $bla_{pAmpC}$ | FOX-M-F<br>FOX-M-R | AACATGGGGTATCAGGGAGATG<br>CAAAGCGCGTAACCGGATTGG | 64 | 190 | [33] |

the recipient solution were mixed, 4 mL of fresh LB broth was added, and the mixture was incubated at 37°C for 16–18 hours. A 0.1-mL aliquot of the cultured cells was spotted and evenly spread on the agar surface of MH agar supplemented with sodium azide (150 mg/L) and cefotaxime (2 mg/L) [34]. If transconjugant colonies were observed, a lysate was prepared from the colony to serve as a DNA template. PCR was performed on the transconjugant strains

**Table 2. Primers used to detect virulence genes.**

| PCR target | Primer | Sequence (5'-3') | Annealing Tm (°C) | Predicted PCR size (bp) | Reference |
|---|---|---|---|---|---|
| rmpA | rmpA-F<br>rmpA-R | ACTGGGCTACCTCTGCTTCA<br>CTTGCATGAGCCATCTTTCA | 55 | 536 | [37] |
| allS | allS-F<br>allS-R | CCGAAACATTACGCACCTTT<br>ATCACGAAGAGCCAGGTCAC | 55 | 508 | [35] |
| wabG | wabG-F<br>wabG-R | ACCATCGGCCATTTGATAGA<br>CGGACTGGCAGATCCATATC | 60 | 683 | [37] |
| uge | uge-F<br>uge-R | TCTTCACGCCTTCCTTCACT<br>GATCATCCGGTCTCCCTGTA | 60 | 534 | [37] |
| entB | entB-F<br>entB-R | ATTTCCTCAACTTCTGGGGC<br>AGCATCGGTGGCGGTGGTCA | 60 | 371 | [36] |
| iutA | iutA-F<br>iutA-R | GGCTGGACATCATGGGAACTGG<br>CGTCGGGAACGGGTAGAATCG | 58 | 300 | [36] |
| kfu | kfu-F<br>kfu-R | ATAGTAGGCGAGCACCGAGA<br>AGAACCTTCCTCGCTGAACA | 60 | 520 | [35] |
| mrkD | mrkD-F<br>mrkD-R | AAGCTATCGCTGTACTTCCGGCA<br>GGCGTTGGCGCTCAGATAGG | 60 | 340 | [16] |
| fimH | fimH-F<br>fimH-R | TGCTGCTGGGCTGGTCGATG<br>GGGAGGGTGACGGTGACATC | 60 | 550 | [35] |

using primers specific for the antibiotic resistance gene present in the donor *K. pneumoniae* strain to confirm the transfer of antibiotic resistance genes.

### Detection of virulence genes

The following virulence genes of *K. pneumoniae* were detected through PCR: fimbriae (*fimH*: type 1 fimbriae, *mrkD*: type 3 fimbriae) [16, 35], iron uptake (*entB*: iron-chelating agent, *iutA*: iron-chelating agent, *kfu*: iron transport and phosphotransferase function) [35, 36], lipopoly-saccharides (*wabG*: synthesis of the lipopolysaccharide core, *uge*: UDP-galacturonate-4-epim-erase) [35], capsule (*rmpA*: a regulator of mucoid phenotype A) [37], and metabolism (*allS*: allantoin metabolism) [35]. The primers used, annealing temperature, and the predicted sizes of PCR products are listed in Table 2.

### Hypermucoviscosity string test

A string test was performed on *K. pneumoniae* isolates inoculated on MHA and incubated at 37˚C overnight. The test involved stretching the mucoid thread of the colony by using a loop, and a positive string test was defined as a stretch greater than 5 mm [38].

### Multilocus sequence typing (MLST)

To understand the prevalence and evolutionary trends of *K. pneumoniae* included in this study, we examined the relationship between the strains through MLST. Seven housekeeping genes of *K. pneumoniae* were amplified through PCR following the method reported by Dia-ncourt et al. [39]. These 7 genes were *rpoB* (a beta subunit of RNA polymerase), *gapA* (glyceral-dehyde 3-phosphate dehydrogenase), *mdh* (malate dehydrogenase), *pgi* (phosphoglucose isomerase), *phoE* (phosphorine E), *infB* (translation initiation factor 2), and *tonB* (periplasmic energy transducer) [39]. The PCR reaction conditions were as follows: initial denaturation at 95˚C for 2 min, followed by 35 cycles of denaturation at 94˚C for 20 s, annealing at 50˚C (*gapA* 60˚C, *tonB* 45˚C) for 30 s, extension at 72˚C for 30 s, and finally extension at 72˚C for 5 min. The reaction was maintained at 4˚C. The primers used are listed in Table 3. The product was electrophoresed to confirm the results of gene amplification. If a PCR product of the cor-rect molecular weight was observed, DNA was excised from the gel by using a blade, placed in a microcentrifuge tube, and sequenced. The sequencing results were uploaded to the Institut Pasteur *K. pneumoniae* MLST database (https://bigsdb.pasteur.fr/klebsiella/) for comparison and sequence type (ST) identification. The similarities between these ESBL/pAmpC-producing *K. pneumoniae* strains were analyzed using BioNumerics version 7.0 (Applied Maths, Sint-Martens-Latem, Belgium).

## Results

### Occurrence of *K. pneumoniae* carrying ESBL and/or pAmpC genes

The 70 *Klebsiella pneumoniae* isolates used in this study (52 from dogs and 18 from cats) were obtained from the urine samples of 52 dogs and 18 cats at National Taiwan University Veteri-nary Hospital between 2014 and 2019. These 70 isolates were initially screened with CHRO-Magar ESBL. Then, 39 of the 70 isolates that exhibited dark blue colonies on the agar surface were tested for ESBL-producing *K. pneumoniae* by using the combination disc test as a pheno-typic confirmation method. From the results of this test, 21 isolates were assumed to produce ESBL. However, sequencing results from the PCR amplification of ESBL *bla* genes indicated that only 19 isolates expressed ESBL *bla* genes. In addition, the PCR of pAmpC genes revealed that 56 of the 70 isolates carried pAmpC *bla* genes. Furthermore, 13 *K. pneumoniae* (12 from

**Table 3. Primers used for *K. pneumoniae* MLST analysis.**

| PCR target | Primer | Sequence (5'-3') | Annealing Tm (°C) | Predicted PCR size (bp) | Reference |
|---|---|---|---|---|---|
| *rpoB* | VIC3<br>VIC2 | GGCGAAATGGCWGAGAACCA<br>GAGTCTTCGAAGTTGTAACC | 50 | 501 | [39] |
| *gapA* | gapA173<br>gapA181 | TGAAATATGACTCCACTCACGG<br>CTTCAGAAGCGGCTTTGATGGCTT | 60 | 450 | [39] |
| *mdh* | mdh130<br>mdh867 | CCCAACTCGCTTCAGGTTCAG<br>CCGTTTTTCCCCAGCAGCAG | 50 | 477 | [39] |
| *pgi* | pgi1F<br>pgi1R<br>pgi2F (seq)<br>pgi2R (seq) | GAGAAAAACCTGCCTGTACTGCTGGC<br>CGCGCCACGCTTTATAGCGGTTAAT<br>CTGCTGGCGCTGATCGGCAT<br>TTATAGCGGTTAATCAGGCCGT | 50 | 432 | [39] |
| *phoE* | phoE604.1<br>phoE604.2 | ACCTACCGCAACACCGACTTCTTCGG<br>TGATCAGAACTGGTAGGTGAT | 50 | 420 | [39] |
| *infB* | infB1F<br>infB1R<br>infB2F (seq) | CTCGCTGCTGGACTATATTCG<br>CGCTTTCAGCTCAAGAACTTC<br>ACTAAGGTTGCCTCCGGCGAAGC | 50 | 318 | [39] |
| *tonB* | tonB1F<br>tonB2R | CTTTATACCTCGGTACATCAGGTT<br>ATTCGCCGGCTGRGCRGAGAG | 45 | 414 | [39] |

dogs and one from cat) did not carry ESBL or pAmpC genes (Table 4). In the 7 ESBL genes that were tested, only $bla_{SHV}$, $bla_{CTX-M-1}$, and $bla_{CTX-M-9}$ groups were detected. Among the 5 pAmpC genes tested, only genes from the $bla_{CIT}$ and $bla_{DHA}$ groups were detected. Table 5 lists the presence of the ESBL and pAmpC genes of the 57 ESBL and/or pAmpC-producing *K. pneumoniae*. In summary, a total of 57 *K. pneumoniae* isolates carried ESBL and/or pAmpC *bla* genes, with one isolate containing only ESBL genes, 38 isolates containing only pAmpC genes, and 18 isolates containing both ESBL and pAmpC genes.

## Antimicrobial susceptibility test

Fig 1 presents the resistance rates of the 57 isolates to various classes of antimicrobial drugs. With the exception of cefotaxime and imipenem, all these isolates exhibited a resistance rate of more than 50%.

## Conjugation test

The transferability of ESBL and pAmpC genes from the *K. pneumoniae* isolates to the *E. coli* J53 strain was evaluated by performing a bacterial conjugation test. Sixteen strain that were naturally resistant to sodium azide was excluded from testing. Of the remaining isolates tested, 43.9% (18/41) could successfully transfer ESBL and/or pAmpC genes to the *E. coli* J53 strain. Among these 18 isolates, 15 contained both ESBL and pAmpC genes, whereas the remaining 3 isolates carried only pAmpC genes. The PCR results revealed that all genes from the $bla_{CTX-M-1}$ and $bla_{CTX-M-9}$ groups were transferred to *E. coli* J53, whereas the $bla_{CIT}$ genes of 3 *K. pneumoniae* isolates (isolate number 2555, 2561, and 2851) were not transferred. A comparison between the original isolate and transconjugant strain is presented in Table 6.

**Table 4. Occurrence of ESBL- and pAmpC-producing *K. pneumoniae* from dog and cat urine samples.**

| species | No ESBL or pAmpC | ESBL | pAmpC | ESBL+pAmpC | ESBL and/or pAmpC |
|---|---|---|---|---|---|
| dog | 11 | 0 | 31 | 10 | 41 |
| cat | 2 | 1 | 7 | 8 | 16 |
| total | 13 | 1 | 38 | 18 | 57 |

**Table 5. Occurrence of ESBL/pAmpC genes from the 57 ESBL and/or pAmpC-producing *K. pneumoniae* isolates.**

| ESBL gene | | | |
|---|---|---|---|
| $bla_{\text{SHV group}}$ | $bla_{\text{CTX-M-1 group}}$ | $bla_{\text{CTX-M-9 group}}$ | $bla_{\text{TEM group}}$, $bla_{\text{CTX-M-2 group}}$, $bla_{\text{CTX-M-8 group}}$, $bla_{\text{CTX-M-25 group}}$ |
| $bla_{\text{SHV-27}}$ (n = 1, 1.8%) | $bla_{\text{CTX-M-3}}$ (n = 8, 14.0%) $bla_{\text{CTX-M-238}}$ (n = 7, 12.3%) | $bla_{\text{CTX-M-14}}$ (n = 4, 7.0%) | Not detected |
| pAmpC gene | | | |
| $bla_{\text{CIT group}}$ | $bla_{\text{DHA group}}$ | | $bla_{\text{MOX group}}$, $bla_{\text{CMY group}}$, $bla_{\text{EBC group}}$, $bla_{\text{FOX group}}$ |
| $bla_{\text{CMY-106}}$ (n = 11, 19.2%) | $bla_{\text{DHA-26}}$ (n = 42, 73.7%) | | |
| $bla_{\text{CMY-174}}$ (n = 2, 3.5%) | | | Not detected |
| $bla_{\text{ACT-53}}$ (n = 16, 28.1%) | | | |
| $bla_{\text{ACT-88}}$ (n = 1, 1.8%) | | | |

## Detection of virulence genes

The prevalence of virulence genes in *K. pneumoniae* was determined using PCR. The genes *wabG* and *entB* were detected in all the 57 isolates. The *mrkD* and *fimH* genes were detected in 98.2% (56/57) of the isolates, whereas the *uge* gene was present in 96.5% (55/57) of the isolates. However, the *kfu* gene was detected in only 33.3% (19/57) of the isolates, and the *iutA* gene was present in only 3.5% (2/57) of the isolates. The *rmpA* and *allS* genes were identified in only 1.8% (1/57) of the isolates each. The most commonly detected virulence gene combination was *wabG*, *uge*, *entB*, *mrkD*, and *fimH* (61.4%, 35/57), followed by *wabG*, *uge*, *entB*, *kfu*, *mrkD*, and *fimH* (31.6%, 18/57). Other combinations were observed only once. Table 7 lists the frequency of the virulence gene combinations.

## Hypermucoviscosity string test

The string test was positive for two *K. pneumoniae* isolates. Fig 2 presents the result of isolate number 2517 that exhibited positivity in the string test.

## Phylogenetic analysis of *K. pneumoniae* isolates

A total of 26 STs were identified among the 57 ESBL- and/or pAmpC-producing *K. pneumoniae* isolates: ST11 (n = 7), ST15 (n = 6), ST655 (n = 6), ST485 (n = 5), ST37 (n = 3), ST3393 (n = 3), ST1825 (n = 3), ST709 (n = 2), ST147 (n = 2), ST967 (n = 1), ST1995 (n = 1), ST846 (n = 1), ST265 (n = 1), ST273 (n = 1), ST592 (n = 1), ST469 (n = 1), ST29 (n = 1), ST966 (n = 1), ST950 (n = 1), ST45 (n = 1), ST198 (n = 1), ST2643 (n = 1), ST1431 (n = 1), ST3216 (n = 1), ST636 (n = 1), ST2817 (n = 1), and unknown STs (n = 3). Among the 7 ST11 isolates, 42.9% (3/7) were obtained from cats and 57.1% (4/7) from dogs. All the six ST15 and ST655 isolates were obtained only from dogs and cats, respectively. Fig 3 displays a visual representation of the MLST results of the 57 isolates, illustrating the relatedness of each ST based on the degree of allele sharing between strains.

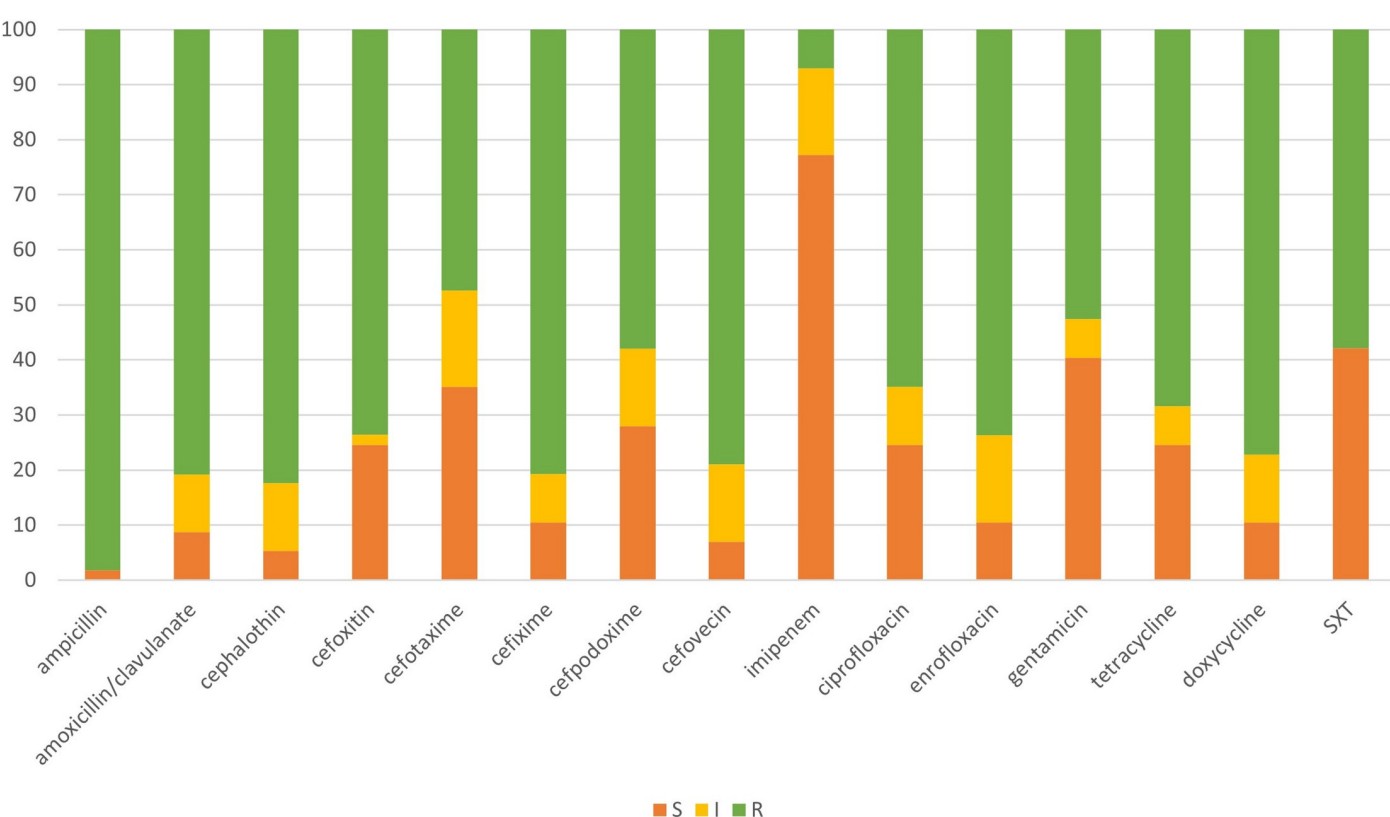

**Fig 1. Antimicrobial susceptibility testing of 57 *K. pneumoniae* isolates harboring ESBL and/or pAmpC genes.** Orange, yellow, and green color bars indicate susceptible (S), intermediate (I), and resistant (R), respectively. The numbers on the Y-axis represent the resistant percentage resistant, and the antimicrobials tested are listed on the X-axis.

**Table 6. *bla* genes detected in donor and the transconjugant strains in the conjugation test.**

| Isolate number | *bla* gene in donor | *bla* in transconjugant |
|:---:|:---|:---|
| 1917 | $bla_{CTX-M-3}$, $bla_{DHA-26}$ | $bla_{CTX-M-3}$, $bla_{DHA-26}$ |
| 2294 | $bla_{CTX-M-3}$, $bla_{CTX-M-14}$, $bla_{DHA-26}$ | $bla_{CTX-M-3}$, $bla_{CTX-M-14}$, $bla_{DHA-26}$ |
| 2544 | $bla_{CTX-M-14}$, $bla_{CMY-106}$, $bla_{DHA-26}$ | $bla_{CTX-M-14}$, $bla_{CMY-106}$, $bla_{DHA-26}$ |
| 2555 | $bla_{CTX-M-238}$, $bla_{ACT-53}$, $bla_{DHA-26}$ | $bla_{CTX-M-238}$, $bla_{DHA-26}$ |
| 2561 | $bla_{CTX-M-14}$, $bla_{CMY-106}$, $bla_{DHA-26}$ | $bla_{CTX-M-14}$, $bla_{DHA-26}$ |
| 2750 | $bla_{CTX-M-14}$, $bla_{CMY-106}$, $bla_{DHA-26}$ | $bla_{CTX-M-14}$, $bla_{CMY-106}$, $bla_{DHA-26}$ |
| 2814 | $bla_{CMY-106}$, $bla_{DHA-26}$ | $bla_{CMY-106}$, $bla_{DHA-26}$ |
| 2837 | $bla_{CTX-M-238}$, $bla_{CMY-174}$ | $bla_{CTX-M-238}$, $bla_{CMY-174}$ |
| 2851 | $bla_{CMY-106}$ | Not detected |
| 2877 | $bla_{CTX-M-238}$, $bla_{DHA-26}$ | $bla_{CTX-M-238}$, $bla_{DHA-26}$ |
| 2899 | $bla_{CTX-M-238}$, $bla_{DHA-26}$ | $bla_{CTX-M-238}$, $bla_{DHA-26}$ |
| 2900 | $bla_{CTX-M-238}$, $bla_{CMY-106}$ | $bla_{CTX-M-238}$, $bla_{CMY-106}$ |
| 2903 | $bla_{CTX-M-3}$, $bla_{DHA-26}$ | $bla_{CTX-M-3}$, $bla_{DHA-26}$ |
| 2904 | $bla_{CTX-M-3}$, $bla_{DHA-26}$ | $bla_{CTX-M-3}$, $bla_{DHA-26}$ |
| 2937 | $bla_{CTX-M-3}$, $bla_{DHA-26}$ | $bla_{CTX-M-3}$, $bla_{DHA-26}$ |
| 2938 | $bla_{CTX-M-3}$, $bla_{DHA-26}$ | $bla_{CTX-M-3}$, $bla_{DHA-26}$ |
| 2953 | $bla_{ACT-53}$, $bla_{DHA-26}$ | $bla_{ACT-53}$, $bla_{DHA-26}$ |
| 2956 | $bla_{CTX-M-238}$, $bla_{DHA-26}$ | $bla_{CTX-M-238}$, $bla_{DHA-26}$ |

**Table 7. Frequencies of virulence gene combinations among 57 ESBL- and/or pAmpC-producing *K. pneumoniae* isolates.**

| Virulence gene combinations | occurrence |
|---|---|
| *allS, wabG, uge, entB, kfu, mrkD, fimH* | 1 |
| *rmpA, wabG, entB, iutA, mrkD, fimH* | 1 |
| *wabG, entB, mrkD, fimH* | 1 |
| *wabG, uge, entB, iutA,* | 1 |
| *wabG, uge, entB, kfu, mrkD, fimH* | 18 |
| *wabG, uge, entB, mrkD, fimH* | 35 |

## Discussion

In this study, we observed a lower prevalence of ESBL genes alone at 1.8% (1/57) than in East Asian countries such as China (75%, 15/20) [40], Japan (82.9%, 29/35) [22], and Korea (42.8%, 12/28) [41]. However, the prevalence of pAmpC genes was significantly higher in our study. We detected pAmpC genes in 66.7% (38/57) of the isolates examined in this study. However, these genes were present in only 10% (2/20) and 11.4% (4/35) of isolates examined in China and Japan, respectively [22, 40]. In Korea, none of the 28 isolates contained pAmpC genes [41]. Additionally, 31.6% (18/57) of the isolates examined in this study harbored both ESBL and pAmpC genes. This proportion was higher than that observed in China (15%, 3/20) and Japan (5.7%, 2/35) but lower than that in Korea (57.1%, 16/28) [22, 40, 41]. In the aforementioned previous studies and the present study, CTX-M type ESBLs, particularly those in the $bla_{CTX-M-1}$ and $bla_{CTX-M-9}$ groups, were the most prevalent ESBLs. Similarly, $bla_{DHA}$ was the dominant gene among pAmpC-containing isolates. However, in our study, one SHV- and no TEM-type ESBLs were detected. Although SHV and TEM type β-lactamases were less common than CTX-M type ESBLs in other studies, most of the SHV and all the TEM genes detected in our study were β-lactamases, not ESBLs. CTX-M type ESBLs are widely prevalent in both humans and companion animals. Among *K. pneumoniae* populations, $bla_{CTX-M-1}$ group enzymes are predominantly found in African and European populations, whereas $bla_{CTX-M-9}$ group enzymes are more common in Asia [42]. In the Taiwan Surveillance of Antibiotic Resistance (TSAR) study conducted between 2002 and 2012, clinical isolates were collected from 25–28 hospitals and medical centers across Taiwan [43]. These isolates were obtained from patients of various age groups and included blood and urine samples. Among the 138 aztreonam-, ceftazidime-, and cefotaxime-resistant isolates analyzed in the study, 54 (39.1%) contained only ESBL genes, 34 (24.6%) contained only AmpC genes, and 27 (19.6%) isolates contained both ESBL and AmpC genes. Within ESBL genes, $bla_{CTX-M}$ genes were predominant (52/81, 64%), followed by $bla_{SHV}$ (24/81, 30%) and $bla_{TEM}$ genes (5/81, 6%). The AmpC genes were predominantly represented by $bla_{DHA}$ (55/61, 90%), a finding consistent with that of the present study.

A significant increasing trend in the prevalence of both ESBL and AmpC genes in *K. pneumoniae* isolates was observed from 2002 to 2012. The detection rate of AmpC-producing isolates increased from 0% to 9.5%, and the incidence of ESBL producers increased from 4.8% to 11.9% [43]. Comparing the results of the TSAR survey, we determined that the prevalence of isolates containing only pAmpC genes or both ESBL and pAmpC genes was higher in isolates obtained from NTUVH than those from human hospitals. However, a significantly lower number of pure ESBL isolates were detected in dogs and cats at NTUVH compared with human patients in the nationwide survey. Previous investigations have reported that the $bla_{CTX-M-1}$, $bla_{CTX-M-9}$, and $bla_{DHA}$ groups of enzymes were distributed across Taiwan and represented the predominant enzymatic groups among *K. pneumoniae* isolates; this finding is

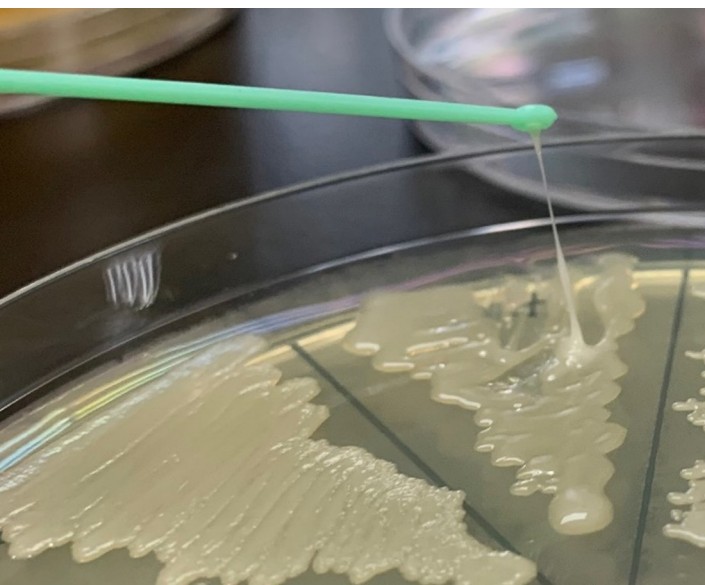

**Fig 2. Hypermucoviscosity string test.** The *K. pneumoniae* isolate number 2517 can be pulled into threads longer than 5 mm, indicating a positive result for the hypermucoviscosity thread test.

in line with that of this study [44]. The consistency in the key *bla* genes identified in *K. pneumoniae* isolated from both humans and animals suggests the potential for the interspecies transmission of *K. pneumoniae* carrying these genes.

Among the 57 isolates analyzed, 26 STs were identified. The most frequently observed ST was ST11, accounting for 12.3% (7/57) of the isolates. Furthermore, ST15, and ST655 were each

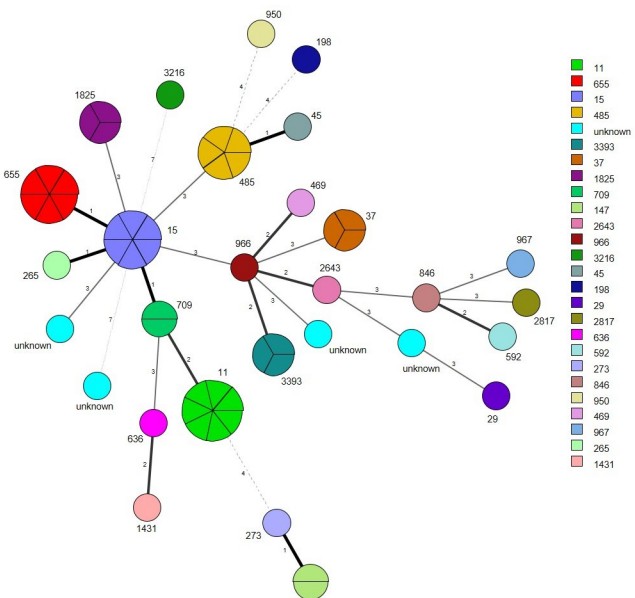

**Fig 3. Minimal spanning tree of ESBL- and/or pAmpC-producing *K. pneumoniae*.** Each circle indicates a sequence type (ST), divided into one sector for each isolate, surrounded by the ST number. The numbers on the connecting line between STs within the MSTree indicate the number of different alleles. Solid lines represent an allelic difference of 3 or less, whereas dotted and faint lines indicate an allelic difference of 4 or more.

detected in 10.5% (6/57) of the isolates, whereas ST485 was detected in 8.8% (5/57) of the samples. Analysis of these STs using the BIGSdb database (https://bigsdb.pasteur.fr/) revealed that all the STs, except for ST265, were originally isolated from human hosts. The source of ST265 remains unknown. ST11, ST15, and ST45, particularly ST11 and ST15, are predominant carbapenem-resistant clones in Asia. These two STs are responsible for up to 60% of carbapenem-resistant *K. pneumoniae* (CRKP) cases in China [45, 46]. Currently, CRKP accounts for a substantial proportion of clinical carbapenem-resistant Enterobacterales (CRE) infections in Europe, China, and the United States, ranging from 60% to 90% [47–49]. In this study, four isolates were identified to be completely resistant to imipenem, a carbapenem antimicrobial, whereas three isolates exhibited intermediate resistance. The resistant isolates were identified as isolate numbers 2899 (ST15), 2900 (ST11), 2937 (ST655), and 2956 (unknown ST). The strains exhibiting intermediate resistance were isolate numbers 2544 (ST11), 2561 (ST11), and 2755 (ST15). These findings are consistent with those of previous surveys, indicating that the majority of *K. pneumoniae* isolates with varying levels of carbapenem resistance are predominantly associated with ST11 and ST15 clones. Carbapenem resistance poses a considerable challenge in the field of infectious disease treatment because carbapenems are typically considered as the final line of defense against severe bacterial infections. Thus, caution should be exercised when CRKP is observed in companion animals. Additionally, a crucial *K. pneumoniae* clone identified in this study is ST147, which accounted for 3.5% (2/57) of the isolates. ST147 is distributed globally and has been reported on all continents, except for Antarctica [50]. Moreover, this clone has been associated with the leading cause of nosocomial outbreaks worldwide [51]. Previous research has demonstrated the widespread presence of ST15 *K. pneumoniae* in companion animals, whereas ST11 is recognized as a high-risk clonal lineage prevalent in human nosocomial infections [52]. However, ST15 was also detected in human hospital and community populations in Portugal [53, 54]. Similarly, ST11 was detected in companion animals in Taiwan, Japan, and Switzerland, with ST11 displaying the highest prevalence among clinical *K. pneumoniae* isolates in Taiwan [23, 43, 55, 56]. These findings provide evidence of shared STs that can occur in both humans and animals, indicating potential epidemiological connections between companion animals and humans due to their close proximity. However, in Italy and France, ST101 and ST274 were predominant in companion animals, suggesting that the predominant ST in ESBL-producing *K. pneumoniae* varies by country [57, 58].

The antimicrobial susceptibility test revealed multidrug-resistant property of ESBL/pAmpC-producing *K. pneumoniae*. Of the 15 antimicrobials tested in this study, more than 50% of *K. pneumoniae* isolates were resistant to 13 of them, with the exception of cefotaxime and imipenem. *K. pneumoniae* strains have been documented to exhibit widespread resistance to aminoglycosides, fluoroquinolones, cephalosporins and carbapenems [59]. Many mobile AMR genes were found in this microorganism before they spread to other pathogens. *K. pneumoniae* is therefore considered an important amplifier and propagator of clinically important AMR genes [60].

In the conjugation assay, some strains grew on MH agar supplemented with sodium azide, which prevents these strains from serving as donor strains. This phenomenon occurs due to the inability to differentiate such donor bacteria from the successful transconjugant, which should grow on MH agar supplemented with cefotaxime and sodium azide. Sodium azide inhibits bacterial growth by inhibiting *secA* [61]. Some *K. pneumoniae* strains may harbor mutations in *secA* or other genes that confer resistance to sodium azide. The successful rate of conjugation in this study was 43.9%, which is lower than that reported by Xiang et al. [62]. A meta-analysis indicated that decreasing taxonomic relatedness between donor and recipient bacteria is associated with lower conjugation frequencies in liquid mating [63]. The frequency of successful conjugations below 50% observed in this study can be explained by the use of the

liquid mating method and the donor and recipient cells belonging to different genera (*K. pneumonia* vs *E. coli*). In addition, temperature and plasmid incompatibility contribute to conjugation frequency [63].

The *wabG* and *entB* genes were identified as the most frequently detected virulence genes. Moreover, *mrkD*, *uge*, and *fimH* genes were highly prevalent in the isolates. The abundance of the *entB* gene in *K. pneumoniae* isolates was expected because siderophores play a crucial role in the uptake of iron, contributing to bacterial survival and enterobactin production in all *K. pneumoniae* strains. A study conducted in Spain demonstrated the prevalence of the *uge* gene in the majority of urine isolates of *K. pneumoniae*. However, the strains of *K. pneumoniae* lacking the *uge* gene exhibited a lower likelihood of causing diseases [36, 64–66].

Hypervirulent *K. pneumoniae* (HvKP) is a highly virulent strain that can infect even healthy hosts. The prevalence of hvKP isolates in Taiwan was 27.5% between 2017 and 2019, with an annual incidence of K1 serotype bacteria ranging from 11%–15% across the country [67]. The HvKP strain can be identified using a phenotypic hypermucoviscosity string test. However, studies have identified specific virulence genes that may be associated with this variant of *K. pneumoniae*. Virulence genes, such as *iutA* and *rmpA*, have been identified as potential markers for the hvKP variant [68, 69]. Limited knowledge exists regarding the prevalence of hvKP in companion animals, with few studies conducted in China and one study in Guadeloupe. In China, hvKP was more common in cats (70%, 14/20) than dogs (58.8%, 50/85). By contrast, only 1 of 4 (25%) dog isolates in the Guadeloupe study was identified as a hypervirulent strain [70–72]. The presence of hvKP in our study underscores the potential role of companion animals as reservoirs in the spread of such bacteria.

## Conclusion

This study sheds light on the prevalence and genetic characteristics of *K. pneumoniae* isolates derived from companion animals with urinary tract infections in Taiwan. The high occurrence of ESBL and pAmpC genes among the isolates underscores the urgent need for vigilance in monitoring antimicrobial resistance in veterinary settings. The distribution of prevalent sequence types (ST11, ST15, and ST655) suggests the circulation of specific *K. pneumoniae* strains across different hosts, emphasizing the interconnectedness of animal and human health. Further research is warranted to elucidate the dynamics of *K. pneumoniae* transmission and resistance dissemination within the One Health framework.

## Supporting information

**S1 Table. The allele number and sequence types (STs) of ESBL and/or pAmpC-producing *K. pneumoniae* analyzed by multilocus sequence typing (MLST).**
(DOCX)

**S2 Table. Presence of the virulence genes of ESBL and/or pAmpC-producing *K. pneumoniae*.**
(DOCX)

**S3 Table. Antimicrobial susceptibility testing of ESBL and/or pAmpC-producing *K. pneumoniae*.**
(DOCX)

## Acknowledgments

The authors extend their gratitude to Dr. Lee-Jene Teng from the Department of Clinical Laboratory Sciences and Medical Biotechnology, National Taiwan University, for generously providing the *Klebsiella pneumoniae* ATCC 700603 strain. Additionally, heartfelt thanks to Dr. Chao-Tsai Liao from the Department of Medical Laboratory Science and Biotechnology, Central Taiwan University of Science and Technology, for kindly provision of the *Escherichia coli* J53 strain.

## Author Contributions

**Conceptualization:** Kuang-Sheng Yeh.

**Funding acquisition:** Kuang-Sheng Yeh.

**Methodology:** Megan Min Yi Lee, Nan-Ling Kuan, Zhi-Yi Li.

**Supervision:** Kuang-Sheng Yeh.

**Writing – original draft:** Megan Min Yi Lee.

**Writing – review & editing:** Kuang-Sheng Yeh.

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
