## [Decision Letter · Decision Letter 0]

2 Nov 2023

PONE-D-23-27932Occurrence and characteristics of extended-spectrum-β-lactamase- and pAmpC-producing Klebsiella pneumoniae isolated from companion animals with urinary tract infectionsPLOS ONE

Dear Dr. Yeh,

Thank you for submitting your manuscript to PLOS ONE. After careful consideration, we feel that it has merit but does not fully meet PLOS ONE’s publication criteria as it currently stands. Therefore, we invite you to submit a revised version of the manuscript that addresses the points raised during the review process.

We look forward to receiving your revised manuscript.

Kind regards,

Nabi Jomehzadeh, Ph.D (Assistant Professor)

Academic Editor

PLOS ONE

Journal Requirements:

This work was supported by a grant (NSTC-112-2313-B-002-031) from the National Science and Technology Council, Taiwan to K.-S. Yeh. The authors extend their gratitude to Dr. Lee-Jene Teng from the Department of Clinical Laboratory Sciences and Medical Biotechnology, National Taiwan University, for generously providing the Klebsiella pneumoniae ATCC 700603 strain. Additionally, heartfelt thanks to Dr. Chao-Tsai Liao from the Department of Medical Laboratory Science and Biotechnology, Central Taiwan University of Science and Technology, for kindly provision of the Escherichia coli J53 strain.

Reviewers' comments:

Reviewer's Responses to Questions

**Comments to the Author**

1. Is the manuscript technically sound, and do the data support the conclusions?

Reviewer #1: Yes

Reviewer #2: Yes

2. Has the statistical analysis been performed appropriately and rigorously? 

Reviewer #1: N/A

Reviewer #2: Yes

3. Have the authors made all data underlying the findings in their manuscript fully available?

Reviewer #1: Yes

Reviewer #2: Yes

4. Is the manuscript presented in an intelligible fashion and written in standard English?

Reviewer #1: Yes

Reviewer #2: Yes

5. Review Comments to the Author

Reviewer #1: 1. The article is well written. The abstract is ok and well-described.

2. The introduction part requires to describe the pathogenic properties of K. pneumoniae a bit stating the important genes and their importance.

3. The materials and methods are well described. A few techniques (like detection of the bacterium by Vitek-2 CMD System) require to be supported by suitable references.

4. Detection of the virulence genes in K. pneumoniae isolates should be discussed in the introduction part too.

5. The results are well written but in complex manner. This part should be properly edited to make the results easily understandable with inclusion of more tables or graphs.

6. The discussion part is well written. All the parameters are well discussed. Comparative analysis with other studies are ok. Few more references supporting the results can be included to support the resistance properties of the ESBL producing K pneumoniae isolates.

7. The conjugational assay and the virulence gene detections are also lacking proper supporting references. Slight elaborate discussion of these parts will make the article more acceptable.

8. The conclusion part is well written.

9. The references are ok and must be in accordance with the journal format.

10. The figures and tables are ok and well described.

Reviewer #2: 1- Line 39, write ‘Klebsiella pneumoniae is a Gram-negative, nonmotile, encapsulated bacterium.’ In all the manuscript write ‘Gram’ not ‘gram’.

2- Line 50, write ‘..ESBL genes are mainly located on plasmids of various molecular sizes, which enhance their horizontal transfer; in addition, those plasmids can confer resistance to other antibiotics such as fluoroquinolones and aminoglycosides [10].’

3- Line 63, write ‘….causing various diseases, such as liver abscess, endophthalmitis,…’

4- Line 169, write: The disc agar diffusion method was used for antimicrobial susceptibility testing [23], and the selected drug discs were as follows: amoxicillin-clavulanic acid (20/10 μg), 171 ampicillin (10 μg), augmentin (5 μg), cefixime (5 μg), cefotaxime (30 μg), cefovecin (30 μg), cefoxitin (30 μg), cefpodoxime (10 μg), cephalothin (30 μg), ciprofloxacin (30 μg), doxycycline (30 μg), enrofloxacin (5 μg), gentamicin (10 μg), imipenem (10 μg), tetracycline (30 μg), and trimethoprim-sulfamethoxazole (25 μg).’

5- In materials and methods section , please indicate the number of isolates that you tested to search the transferability of ESBL/pAmpC transfer. Since it is not clear you tested only one train or many strains, readers realize that you tested 41 isolates only in section results. You can say that only sodium azid resistant isolates were tested for conjugaison; you said this in section results but it is better to say that in section MM.

6- In the beginning of the section discussion you spook about results reported previously in china, Japan (line 337), but you do not provide reference articles of these data, please add the references.

7- In section discussion, generally K. pneumonia contains the SHV-1 (according to literature) so how you explain the absence of this gene in the majority of your strains, provide one or two sentences to explain this.

8- Line 376 you said ‘ST11, accounting for 12.3% (7/57)’ but in line 377 you said ‘.Furthermore, ST11, ST15, and ST655 were detected in 10.5% (6/57) of the isolates each..’ please verify the frequency of ST11 is 12.3% or 10.5%.

6. PLOS authors have the option to publish the peer review history of their article (what does this mean?). If published, this will include your full peer review and any attached files.

Reviewer #1: No

Reviewer #2: **Yes: **Mohamed Salah Abbassi

---

## [Author Response · Author response to Decision Letter 0]

18 Nov 2023

Editor

PLOS ONE

Dear Editor:

I list my responses to the comments raised by reviewer #1 and reviewer #2 as follows:

Reviewer #1: 

1. The article is well written. The abstract is ok and well-described.

Response: Thanks.

2. The introduction part requires to describe the pathogenic properties of K. pneumoniae a bit stating the important genes and their importance.

Response: Lines57-72, a paragraph has been added that briefly describes the pathogenic properties of K. pneumoniae and important virulence factors. Relevant references have also been provided. 

3. The materials and methods are well described. A few techniques (like detection of the bacterium by Vitek-2 CMD System) require to be supported by suitable references.

Response: The Vitek 2 system is an automatic microbial identification facility based on microbial biochemical testing. A new reference entitled “Evaluation of the VITEK 2 system for rapid identification of medically relevant Gram-negative rods” published in the Journal of Clinical Microbiology” (DOI: https://doi.org/10.1128/jcm.36.7.1948-1952.1998), was added (reference 25).

4. Detection of the virulence genes in K. pneumoniae isolates should be discussed in the introduction part too.

Response: please refer to response to question 2. At this part, brief description of virulence genes was added. A citation reference 16 that describes the PCR method to detect virulence genes of K. pneumoniae was added.

5. The results are well written but in complex manner. This part should be properly edited to make the results easily understandable with inclusion of more tables or graphs.

Response: An additional table, Table 5, was provided, the contents of which would, to some extent, replace the text describing the presence of the ESBL and pAmpC genes in the 57 ESBL- and/or pAmpC-producing K. pneumoniae in the original manuscript (lines 263-267). The relevant content in the text has therefore been removed.

6. The discussion part is well written. All the parameters are well discussed. Comparative analysis with other studies are ok. Few more references supporting the results can be included to support the resistance properties of the ESBL producing K pneumoniae isolates.

Response: lines 423-431, A paragraph briefly reporting our antimicrobial susceptibility result and two additional citations (reference 59 and 60) were added. We hope that this part can highlight the resistance characteristics of ESBL/pAmpC-producing K. pneumoniae in the present study. 

7. The conjugational assay and the virulence gene detections are also lacking proper supporting references. Slight elaborate discussion of these parts will make the article more acceptable.

Response: Line 195 and 205, a citation for conjugation test reference 34 has been added here. In lines 213-218, citations for each virulence gene detection were added. The same quotes also appear in Table 2.

8. The conclusion part is well written.

Response: thanks

9. The references are ok and must be in accordance with the journal format.

Response: I will make sure the reference style matches the journal format.

10. The figures and tables are ok and well described.

Response: thanks

Reviewer #2: 

1. Line 39, write ‘Klebsiella pneumoniae is a Gram-negative, nonmotile, encapsulated bacterium.’ In all the manuscript write ‘Gram’ not ‘gram’.

Response: line37, gram was replaced by Gram.

2. Line 50, write ‘..ESBL genes are mainly located on plasmids of various molecular sizes, which enhance their horizontal transfer; in addition, those plasmids can confer resistance to other antibiotics such as fluoroquinolones and aminoglycosides [10].’

Response: line 48-50, the original sentence has been changed accordingly.

3. Line 63, write ‘….causing various diseases, such as liver abscess, endophthalmitis,…’

Response: line 78, the word conditions was replaced with diseases.

4. Line 169, write: The disc agar diffusion method was used for antimicrobial susceptibility testing [23], and the selected drug discs were as follows: amoxicillin-clavulanic acid (20/10 μg), 171 ampicillin (10 μg), augmentin (5 μg), cefixime (5 μg), cefotaxime (30 μg), cefovecin (30 μg), cefoxitin (30 μg), cefpodoxime (10 μg), cephalothin (30 μg), ciprofloxacin (30 μg), doxycycline (30 μg), enrofloxacin (5 μg), gentamicin (10 μg), imipenem (10 μg), tetracycline (30 μg), and trimethoprim-sulfamethoxazole (25 μg).’

Response: line 178-180, the original content has been replaced accordingly.

5. In materials and methods section, please indicate the number of isolates that you tested to search the transferability of ESBL/pAmpC transfer. Since it is not clear you tested only one strain or many strains, readers realize that you tested 41 isolates only in section results. You can say that only sodium azid resistant isolates were tested for conjugaison; you said this in section results but it is better to say that in section MM.

Response: line 186-190, in the Materials and methods section, a statement “Since 16 of the total 57 ESBL/pAmpC-producing K. pneumoniae strains were naturally resistant to sodium azide because they could grow on Mueller–Hinton agar supplemented with sodium azide for unknown reason; Therefore, these strains were excluded from the conjugation test. The number of strains used for the conjugation test was 41” was added. The similar statement in the original Results section was then removed.

6. In the beginning of the section discussion you spook about results reported previously in china, Japan (line 337), but you do not provide reference articles of these data, please add the references.

Response: Lines343-351, Corresponding references have been placed after the corresponding statements.

7. In section discussion, generally K. pneumonia contains the SHV-1 (according to literature) so how you explain the absence of this gene in the majority of your strains, provide one or two sentences to explain this.

Response: blaSHV-1 was first identified in 1970s in E. coli. The encoded enzyme SHV-1 exhibits activity against penicillins and first generation of cephalosporins. SHV-1 is not an ESBL. In fact, we were able to detect 20 SHVs from our K. pneumoniae strains, but with the exception of SHV-27 being an ESBL type, the others were not ESBLs. Therefore, we did not include them as ESBL SHV-producing K. pneumoniae. In lines 354-357, I wrote: “However, in our study, one SHV- and no TEM-type ESBLs were detected. Although SHV and TEM type β-lactamases were less common than CTX-M type ESBLs in other studies, most of the SHV and all the TEM genes detected in our study were β-lactamases, not ESBLs”. I hope this can clear readers’ confusion.

8. Line 376 you said ‘ST11, accounting for 12.3% (7/57)’ but in line 377 you said ‘Furthermore, ST11, ST15, and ST655 were detected in 10.5% (6/57) of the isolates each.’ please verify the frequency of ST11 is 12.3% or 10.5%.

Response: This was an error in the original manuscript. ST11 has been removed from the original text. In this revised manuscript, lines 388-389, ST15 and ST655 were each detected in 10.5% (6/57) of the isolate… 

Regards

Yours

Kuang-Sheng Yeh, DVM, PhD

Department of Veterinary Medicine

School of Veterinary Medicine

National Taiwan University, Taipei, Taiwan

Tel: 886-2-33661289

E-mail: ksyeh@ntu.edu.tw

---

## [Decision Letter · Decision Letter 1]

18 Dec 2023

Occurrence and characteristics of extended-spectrum-β-lactamase- and pAmpC-producing Klebsiella pneumoniae isolated from companion animals with urinary tract infections

PONE-D-23-27932R1

Dear Dr. Yeh,

We’re pleased to inform you that your manuscript has been judged scientifically suitable for publication and will be formally accepted for publication once it meets all outstanding technical requirements.

Kind regards,

Nabi Jomehzadeh, Ph.D (Assistant Professor)

Academic Editor

PLOS ONE

Additional Editor Comments (optional):

Reviewers' comments:

Reviewer's Responses to Questions

**Comments to the Author**

1. If the authors have adequately addressed your comments raised in a previous round of review and you feel that this manuscript is now acceptable for publication, you may indicate that here to bypass the “Comments to the Author” section, enter your conflict of interest statement in the “Confidential to Editor” section, and submit your "Accept" recommendation.

Reviewer #1: All comments have been addressed

2. Is the manuscript technically sound, and do the data support the conclusions?

Reviewer #1: Yes

3. Has the statistical analysis been performed appropriately and rigorously? 

Reviewer #1: Yes

4. Have the authors made all data underlying the findings in their manuscript fully available?

Reviewer #1: Yes

5. Is the manuscript presented in an intelligible fashion and written in standard English?

Reviewer #1: Yes

6. Review Comments to the Author

Reviewer #1: The article is well revised. All the comments and queries are addressed. The revised article is well explained in all the sections from abstract to conclusion.

7. PLOS authors have the option to publish the peer review history of their article (what does this mean?). If published, this will include your full peer review and any attached files.

Reviewer #1: No

---

## [Editor Report · Acceptance letter]

5 Jan 2024

PONE-D-23-27932R1 

PLOS ONE

Dear Dr. Yeh, 

I'm pleased to inform you that your manuscript has been deemed suitable for publication in PLOS ONE. Congratulations! Your manuscript is now being handed over to our production team.

Kind regards, 

on behalf of

Dr. Nabi Jomehzadeh 

Academic Editor

PLOS ONE